# Research Progress on the Mechanism of Bile Acids and Their Receptors in Depression

**DOI:** 10.3390/ijms26094023

**Published:** 2025-04-24

**Authors:** Xue Zhao, Iin Zheng, Wenjing Huang, Dongning Tang, Meidan Zhao, Ruiling Hou, Ying Huang, Yun Shi, Weili Zhu, Shenjun Wang

**Affiliations:** 1Research Center of Experimental Acupuncture Science, Tianjin University of Traditional Chinese Medicine, Tianjin 301617, China; zhaox981122@163.com (X.Z.); zhenglin0592@163.com (I.Z.); hwjysh@163.com (W.H.); tangldn@outlook.com (D.T.); zhaomeidan2012@163.com (M.Z.); hrlhrl11@163.com (R.H.); chianti032703@163.com (Y.H.); 2School of Acupuncture & Moxibustion and Tuina, Tianjin University of Traditional Chinese Medicine, Tianjin 301617, China; 3National Clinical Research Center for Chinese Medicine Acupuncture and Moxibustion, Tianjin 300381, China; 4Hebei Key Laboratory of Early Life Health Promotion, Hebei Medical University, Shijiazhuang 050031, China; shycaoer@163.com; 5National Institute on Drug Dependence and Beijing Key Laboratory of Drug Dependence Research, Peking University, Beijing 100191, China

**Keywords:** bile acids, bile acid receptors, depression, gut–brain axis, inflammation, neuroplasticity

## Abstract

Depression, a highly prevalent mental disorder worldwide, arises from multifaceted interactions involving neurotransmitter imbalances, inflammatory responses, and gut–brain axis dysregulation. Emerging evidence highlights the pivotal role of bile acids (BAs) and their receptors, including farnesoid X receptor (FXR), Takeda G protein-coupled receptor 5 (TGR5), and liver X receptors (LXRs) in depression pathogenesis through modulation of neuroinflammation, gut microbiota homeostasis, and neural plasticity. Clinical investigations demonstrated altered BA profiles in depressed patients, characterized by decreased primary BAs (e.g., chenodeoxycholic acid (CDCA)) and elevated secondary BAs (e.g., lithocholic acid (LCA)), correlating with symptom severity. Preclinical studies revealed that BAs ameliorate depressive-like behaviors via dual mechanisms: direct CNS receptor activation and indirect gut–brain signaling, regulating neuroinflammation, oxidative stress, and BDNF/CREB pathways. However, clinical translation faces challenges including species-specific BA metabolism, receptor signaling complexity, and pharmacological barriers (e.g., limited blood–brain barrier permeability). While FXR/TGR5 agonists exhibit neuroprotective and anti-inflammatory potential, their adverse effects (pruritus, dyslipidemia) require thorough safety evaluation. Future research should integrate multiomics approaches and interdisciplinary strategies to develop personalized BA-targeted therapies, advancing novel treatment paradigms for depression.

## 1. Introduction

The World Health Organization (WHO) estimates that more than 350 million people of all ages suffer from depression worldwide and that it is a significant contributor to the global burden of disease. Currently ranked as the fourth leading cause of disability, WHO projects that it may take over as the leading cause by 2030. Characterized by a persistent low mood and diminished interest or pleasure, the etiology of depression relates to a wide spectrum: monoamine neurotransmitter deficits; dysregulation in the hypothalamic–pituitary–adrenal (HPA) axis; inflammatory responses; impairments in neuroplasticity and neurogenesis; changes in brain structure and function; and genetic influences, environmental stressors, and epigenetic influences. Besides side effect management, depression diagnosis and treatment are great challenges to patients [1,2].

Emerging evidence underscores the crucial role of bile acids and their receptors in various physiological processes and potentially in the therapeutic intervention of depression. They mitigate inflammation through a reduction of inflammatory cytokine production and are associated with the nexus between depression and neuroinflammation. Bile acids also enter into a bidirectional regulatory dynamic with the gut microbiota, thereby influencing brain health and emotional states through the gut–brain axis. Moreover, it is hypothesized that they may enhance neuroplasticity by upregulating neuroprotective factors involved in repairing and strengthening damaged neural circuits, thereby exerting a positive influence on the treatment of depression. Bile acids and their receptors improve other comorbid conditions of depression, including mental disorders, obesity, diabetes, and irritable bowel syndrome [3]. This proves the multifactorial etiology of depression and also supports the use of multimechanism therapeutic strategies. However, research in this field is still in its nascent stages, and there is a pressing need for more comprehensive studies to definitively unravel the underlying mechanisms and explore its therapeutic applications.

## 2. Overview of Bile Acids and Their Receptor Families

### 2.1. Bile Acids

Bile acids (BAs) function as nutrient detergents, dissolving lipids and fat-soluble vitamins. Synthesized in the liver from cholesterol, primary BAs are stored in the gallbladder and are integral to bile composition. Post-prandial secretion into the intestine facilitates their transformation into secondary BAs through gut microbiota-mediated processes. The majority of BAs are reabsorbed by intestinal epithelial cells and returned to the liver via portal circulation, with a minor fraction entering systemic circulation. BAs, due to their chemical structure and lipid solubility, can traverse the blood–brain barrier, either by diffusion or via transport proteins, though direct evidence remains elusive. A minor fraction is excreted in feces.

Structurally, BAs are categorized into free and conjugated forms. Free BAs include cholic acid (CA), deoxycholic acid (DCA), lithocholic acid (LCA), chenodeoxycholic acid (CDCA), and ursodeoxycholic acid (UDCA). Conjugated BAs are formed by the amidation of free BAs with glycine or taurine, yielding compounds such as glycocholic acid (GCA), glycochenodeoxycholic acid (GCDCA), taurocholic acid (TCA), and taurochenodeoxycholic acid (TCDCA).

The canonical BA synthesis pathway occurs predominantly in the liver, where cholesterol is hydroxylated by cytochrome P450 7A1 (CYP7A1) to form 7α-hydroxycholesterol, subsequently processed into primary BAs (CA and CDCA). An alternative pathway mediated by CYP27A1 generates CDCA. Notably, cerebral BA synthesis involves CYP46A1-mediated conversion of cholesterol to CDCA, which is further modified into secondary BAs (DCA, LCA, UDCA) by gut microbiota. Circulating BAs can also traverse the blood–brain barrier (BBB) to access the central nervous system (refer to Figure 1) [4,5]. Importantly, interspecies differences exist: humans primarily synthesize CA and CDCA, whereas rodents produce CA and muricholic acids (MCAs) as major primary BAs [6].

BAs interact with the central nervous system through direct and indirect pathways, activating farnesoid X receptor (FXR) and G-protein-coupled bile acid receptor 5 (TGR5) within the brain, and releasing fibroblast growth factor 19 (FGF19) and glucagon-like peptide-1 (GLP-1) from the intestine to signal the central nervous system [4,5]. Additionally, BAs serve as multifunctional signaling molecules, regulating their synthesis by activating receptors that release intermediate signaling molecules for downstream signaling [7].

### 2.2. Bile Acid Receptors

Bile acid receptors (BARs) are proteins that interact with BAs to mediate biological effects. BARs are classified into nuclear and membrane receptors, with the former regulating gene transcription and expression and the latter sensing and transmitting signals, maintaining cellular homeostasis, and mediating intercellular interactions. The primary nuclear receptor is FXR, while the main membrane receptor is TGR5 [8].

FXR, identified as the first bile acid-activated receptor in 1999, is a nuclear transcription factor and the most extensively studied member of the bile acid nuclear receptor family. Its main endogenous ligand is bile acid, with CDCA exhibiting the highest affinity, followed by DCA, LCA, and CA. FXR is predominantly expressed in the liver, intestine, kidney, adrenal glands, and brain [9]. Activation of FXR regulates the gene expression of various bile acid metabolic enzymes, controlling bile acid synthesis and circulation. FXR also influences lipid metabolism by inhibiting fatty acid synthesis and promoting bile acid pool formation, impacting metabolic diseases such as obesity and fatty liver [10]. Gut microbiota can decrease FXR expression in the ileum, inhibiting bile acid synthesis in the liver. FXR plays a crucial role in regulating gut function and the gut microbiota [11]. In the liver, high bile acid levels lead to FXR binding with the small heterodimer partner (SHP), inhibiting the activation of key enzymes involved in bile acid synthesis through a feedback loop that reduces bile acid synthesis. In the intestine, FXR induces fibroblast growth factor 15 (FGF15; FGF19 in humans) to activate hepatic FGF receptor 4 (FGFR4), thereby inhibiting CYP7A1 [12].

Liver X receptors (LXRs), comprising two isoforms LXRα (NR1H3) and LXRβ (NR1H2), exhibit distinct tissue distributions: LXRα is highly expressed in the liver, intestine, and macrophages, whereas LXRβ is ubiquitously distributed across tissues. LXRs function as cholesterol sensors by binding oxidized cholesterol derivatives (e.g., 24(S)-hydroxycholesterol), which are metabolic intermediates that activate their transcriptional activity, thereby regulating key enzymes in cholesterol metabolism and BAs synthesis. Mechanistically, LXRs bind to and modulate the promoter region of CYP7A1, the rate-limiting enzyme in BA synthesis. Studies demonstrated that LXR activation upregulates CYP7A1 expression, promoting cholesterol conversion to BAs and maintaining systemic cholesterol homeostasis [13,14]. Additionally, LXRs govern hepatocyte cholesterol biosynthesis and metabolism via ABC transporters [15]. Notably, LXRs antagonize FXR signaling but synergistically coordinate BA balance through complementary regulatory pathways.

TGR5 is a membrane receptor expressed in various tissues and cells. LCA has the highest affinity for TGR5, followed by DCA, CDCA, and CA. TGR5 is predominantly expressed in the liver, spleen, intestine, brain, and fat tissues, with high expression levels in macrophages [16,17]. Bile acid activation of peripheral TGR5 can increase energy expenditure in brown fat, promoting heat generation, which influences body weight regulation and metabolic diseases such as obesity and diabetes. TGR5 is considered a critical target for type 2 diabetes and metabolic syndrome [18]. In the hypothalamus, TGR5 can control diet-induced obesity, as bile acid supplements or TGR5 agonists can reduce sympathetic nerve activity to lower weight and treat obesity. Selective activation of TGR5 in the brain effectively reduces food intake [19]. RNA interference experiments have shown that reducing TGR5 expression in enteroendocrine cells in the small intestine significantly lowers bile acid-induced glucagon-like peptide-1 (GLP-1) secretion, while increased TGR5 expression enhances GLP-1 secretion. This indicates that bile acids can promote GLP-1 secretion through TGR5 in enteroendocrine cells of mice, thereby improving blood glucose levels [20].

## 3. The Relationship Between Bile Acids, Their Receptors, and Depression

BA metabolism is significantly associated with depressive symptom severity in MDD. Clinical studies [21,22] reveal a characteristic BA imbalance in MDD patients: serum levels of primary BAs (CA and CDCA) are markedly reduced in severe depression with comorbid anxiety compared to mild cases, while gut microbiota-derived secondary BAs (e.g., LCA) and the LCA/CDCA ratio are abnormally elevated in high-anxiety subgroups. Notably, non-responders to antidepressants exhibit significantly lower baseline CDCA levels and more pronounced dysregulation of the 7-keto-LCA/CDCA ratio than responders. Although current evidence primarily derives from intra-cohort analyses, this primary–secondary BA axis disruption has been pathologically linked to gut–brain axis signaling, particularly FXR pathway activation. Further investigations [23] demonstrated that MDD patients, compared to healthy controls, show not only reduced CDCA levels but also diminished upstream precursors (e.g., 3,7-dihydroxy-5-cholanic acid) and cholesterol intermediates, indicating systemic impairment of primary BA synthesis. Critically, elevated BAs concentrations may compromise BBB integrity, disrupting emotion-related neural circuits [24,25]. This BA metabolic reprogramming likely exacerbates affective symptoms through dual mechanisms: CDCA depletion attenuates FXR-mediated neuroprotection, aggravating neuroinflammation and synaptic dysfunction, while secondary BA accumulation disrupts central neurotransmitter homeostasis via direct or indirect pathways. Collectively, BA metabolic signatures may serve not merely as contributors to MDD pathophysiology but as dynamic biomarkers for predicting therapeutic responses and guiding personalized interventions (refer to Figure 1).

Research indicates that BA concentrations exhibit complex changes in relation to depression, with specific effects potentially influenced by multiple factors. Different BA subtypes and total BA concentrations may have varying implications. Non-targeted metabolomics identified potential biomarkers in the serum of CUMS-induced depressive mice, showing elevated levels of CDCA (345%↑, *p* = 0.0314), DCA (220%↑, *p* = 0.0152), and CA (197%↑, *p* = 0.0009), while TCA was significantly reduced (56%↓, *p* = 0.0452) [26]. Compared with healthy subjects, major depressive disorder (MDD) patients exhibited lower total serum BA levels [27]. However, an increased BA concentration is not uniformly beneficial for the nervous system; excessively high BA levels can have adverse effects, including cytotoxicity. For example, in obstructive cholestasis, elevated BAs in the blood circulation can increase blood–brain barrier permeability, raising the risk of depression [28,29].

Regarding BA receptors, excessive activation of the farnesoid X receptor (FXR) may impact brain neurofunction by influencing neuroplasticity and neurodevelopment, potentially leading to depression. Conversely, FXR activation can positively affect depression by enhancing anti-inflammatory responses, repairing neural damage, and promoting neurogenesis. Therefore, it is imperative to appropriately regulate BA and receptor levels to avoid the potential adverse effects of excess. Additionally, after applying agonists in TGR5 gene-knockout mice, the stress response could not be completely suppressed, indicating other factors may contribute to chronic stress-induced depressive behaviors. In the brain, TGR5 has a neurosteroid receptor function; loss of steroid protection makes the brain more susceptible to cortisol and other toxins, potentially leading to depression-like behaviors. Chronic stress may also influence TGR5 expression through steroids [16,30]. Given the complex variations of BAs and their receptors in depression, the following discussion examines inflammation, gut microbiota, and neuroplasticity, with a research summary provided in Table 1 and Figure 2.

### 3.1. Inflammation

The onset of depression is associated with chronic inflammation and immune system activation. Peripheral inflammation, stemming from infections, trauma, or other diseases, can lead to the activation of immune cells such as macrophages, which release inflammatory factors. These factors can cross the blood–brain barrier, enter the brain, activate glial cells and neurons, and lead to abnormal neuronal activity and neurotransmitter release, thereby affecting brain regions’ functions and physiological processes related to emotion regulation. Studies have found that depressive patients have higher levels of pro-inflammatory biomarkers in their blood [40,41,42]. Recently, some evidence has shown that BAs and their receptors are also implicated in inflammation regulation and immune response, which contributes to anti-depressive effects.

UDCA and its derivatives, GUDCA and TUDCA, have been considered to modulate apoptosis, oxidative stress, and inflammation in the nervous system. These compounds exerted their effects by regulating mitochondrial function, inhibiting the signaling pathways of cell death, reducing oxidative stress, and modulating the release of inflammatory factors. All of these benefits protect against neurodegenerative diseases [43]. Of note, TUDCA improved depression-like behaviors of CUMS mice as shown in the tail suspension test, the forced swim test, and the sucrose preference test. Its mechanism involves normalization of tumor necrosis factor-alpha (TNF-α) and interleukin-6 (IL-6) levels in the hippocampus and the prefrontal cortex; reduction of inflammatory markers, including interleukin-1β (IL-1β); NLRP3 and Iba-1 activation; inhibition of oxidative nitrosative stress; and reduction of endoplasmic reticulum stress markers (nitric oxide, reduced glutathione, malondialdehyde, glucose-regulated protein 78, and C/EBP homologous protein) after 10 days of TUDCA treatment [31]. Moreover, the research on LPS (lipopolysaccharide)-induced depressive model mice showed that the anti-depressive functions of TUDCA are most likely exerted by the mechanisms mentioned above and are more effective when in dosages of 200 mg/kg and 400 mg/kg rather than 100 mg/kg [32]. These findings provide valuable insights for developing new treatment strategies for depression.

In vitro studies have shown that FXR can alleviate inflammation by inhibiting pro-inflammatory gene transcription by transcription factors such as nuclear factor-κB (NF-κB). FXR-knockout mice exhibit severe hepatic inflammation and spontaneous liver tumors [44]. Haoran Zhang et al. [33] found that ganoderic acid A can inhibit NLRP3 inflammasome activity in the prefrontal cortex of a social defeat stress model in mice by regulating FXR, impacting the activation and release of caspase-1 and IL-1β, enhancing AMPA receptor surface stability in the hippocampus, and further promoting glutamatergic neurotransmission in the hippocampal region, ultimately alleviating depression-like behaviors in mice. FXR-knockout mice also showed an increase in tail suspension immobility time and blocked the antidepressant effect of ganoderic acid A. Additionally, ganoderic acid A is considered to have multiple pharmacological activities, including antioxidant, anti-inflammatory, anti-tumor, and neuroprotective effects, and it can be safely combined with antidepressants to increase the likelihood of treatment success.

Preclinical studies in CUMS and corticosterone (CORT)-induced rat models demonstrate that LXRβ activation alleviates depression-like behaviors. Mechanistically, the LXRβ agonist TO901317 suppresses neuroinflammation by inhibiting the NF-κB/NLRP3 pathway in the basolateral amygdala (BLA), reducing microglial pro-inflammatory and phagocytic activity while decreasing IL-1β release [34]. LXRβ activation concurrently modulates BAs homeostasis, attenuating toxic secondary BAs accumulation to reinforce anti-inflammatory effects. Plasma proteomic profiling of CUMS mice via iTRAQ technology identified 47 differentially expressed proteins, with significant dysregulation in the LXR/RXR activation pathway. Key alterations include upregulated acute-phase response proteins (lipopolysaccharide-binding protein, α-1 antitrypsin) and downregulated fibrinogen beta chain, implicating this pathway in neuroinflammatory processes through complement system activation. Aberrant LXR/RXR signaling further correlates with lipid metabolic disturbances, exemplified by apolipoprotein A2 dysregulation impairing cholesterol transport. Functional impairment of this pathway exacerbates depression-associated neuroimmune-metabolic abnormalities by amplifying inflammatory responses (e.g., TLR4 signaling) and disrupting membrane lipid homeostasis [35]. These findings collectively reveal that LXRβ exerts antidepressant effects via neuro-immune microenvironment modulation, with its crosstalk with BA metabolism representing a promising therapeutic target. This work provides an experimental rationale for targeting LXRβ and BA metabolic networks in depression intervention.

TGR5 is also closely associated with inflammation. Laboratory studies have shown that TGR5 can inhibit cytokine production in macrophages by activating the cAMP–NF–κB signaling pathway, reduce inflammation in arterial walls, and promote cholesterol excretion, helping to slow or prevent the progression of atherosclerosis [45]. Moreover, the TGR5 agonist INT-777 significantly reduced neuroinflammation in LPS-induced Alzheimer’s model mice via upregulating TGR5 and inhibiting NF-κB-mediated pro-inflammatory cytokine expression and microglial activation, thereby reducing neuroinflammation and apoptosis and alleviating synaptic dysfunction [46]. Regulating inflammatory signaling pathways and cytokine production has potential therapeutic effects on inflammatory diseases. Therefore, TGR5 may have therapeutic roles in depression-related inflammation, though its specific mechanisms remain unclear.

### 3.2. Gut Microbiota

Gut microbiota, integral to the gut–brain axis, interact with the nervous system and influence brain function and behavior through the production of hormones, metabolites, and neurotransmitters. Certain microbiota can regulate proteins (such as brain-derived neurotrophic factor), neurotransmitters, and their precursors (such as serotonin, gamma-aminobutyric acid, dopamine, and tryptophan), which are closely related to mental health. Dysbiosis in the microbiota can lead to mood disorders and cognitive impairments, with significant differences in gut microbiota composition between healthy individuals and depression patients [47]. BAs, as metabolites of gut microbiota, have signaling abilities and receptor-binding capacities that are modulated by the microbiota, while BAs can also regulate the microbiota in turn. Additionally, BAs act on the brain through both direct and indirect pathways, entering the circulation and promoting information exchange between the gut and brain.

Increasing evidence suggests that BA synthesis and metabolism are associated with depression. Serum metabolomics and fecal metagenomic sequencing were conducted on 104 patients with major depressive disorder (MDD) and 77 healthy controls. For the first time, a multiomics combined analysis method was adopted, and the results were verified in another independent cohort. This comprehensively revealed the inherent connections among the gut microbiota, bile acid metabolism, and cognitive function in depression. Specifically, specific gut microbes related to depression were precisely identified, such as multiple beneficial bacteria in genera like Lachnospiraceae and Ruminococcaceae, and *Escherichia coli*, which were significantly decreased in patients with depression, as well as their regulatory roles in key enzymes within the bile acid metabolism pathway. Through the integration of omics information, a disease classifier based on gut microbiota and bile acid was successfully constructed, which can effectively distinguish patients with depression from healthy individuals, providing new potential biomarkers for the diagnosis of depression [48].

A targeted metabolomic analysis of serum from 208 untreated MDD patients found that BA concentrations were correlated with scores on the 17-item Hamilton Depression Scale and the 14-item Hamilton Anxiety Scale [22]. Compared to healthy individuals, depression patients exhibited higher levels of Enterococcus bacteria in the gut and increased serum levels of 23-nordeoxycholic acid (NorDCA). Taurolithocholic acid (TLCA), glycolithocholic acid (GLCA), and lithocholic acid 3-sulfate (LCA-3S) were negatively correlated with depression scores on the Hamilton Scale and positively correlated with the relative abundance of Turicibacter [49]. Gene sequencing studies on CUMS model mice showed an increase in secondary bile acid DCA in the feces, which was positively associated with three members of the Firmicutes phylum (Ruminococcaceae, Ruminococcus, and Clostridia) [26].

Chaihu-Shugan-San (CSS), a traditional Chinese herbal formula, is known to soothe the liver and regulate Qi to treat emotional disorders. Recent reports indicate that CSS can alter the composition of gut microbiota and related BA metabolites. A study conducted by the Xiangya Medical School of Central South University observed that transplanting gut bacteria from CUMS-treated mice given CSS into stressed mice restored serum levels of secondary BAs—hyocholic acid (HCA) and 7-ketodeoxycholic acid (7-ketoDCA)—alleviating depression-like behaviors. This suggests that gut microbiota play a key role in regulating BA metabolism and diversity, highlighting CSS’s potential antidepressant effects through the modulation of gut microbiota [36]. These findings indicate a correlation between gut microbiota and BA metabolism, with dysregulation in both potentially contributing to the pathogenesis of depression.

Research shows that the vagus nerve serves as an important bridge in gut–brain interactions, helping to maintain internal balance and stability. Gut microbes can modulate chronic stress-induced depression in mice via the afferent pathways of the vagus nerve [50]. Severing the vagus nerve impairs the gut barrier’s tight junctions, elevating serum BA levels [51]. Intestinal TGR5 can promote GLP-1 release from intestinal L-cells, and circulating GLP-1 affects the brain by activating GLP-1 receptors in vagal afferent neurons [52]. GLP-1 receptor agonists have also been described as potential antidepressants [53]. Therefore, gut microbiota may alleviate depressive symptoms through the mediation of the vagus nerve.

### 3.3. Neuroplasticity

Depression is often accompanied by changes in neuroplasticity, which may result in abnormal connectivity and function between neurons, affecting the processing of emotions and cognition. For example, depression patients may exhibit reduced hippocampal plasticity, which is linked to memory and emotional regulation impairments [54]. Preliminary studies have indicated that BAs may be associated with neuroplasticity. UDCA, for instance, acts as a cytoprotective agent that prevents programmed cell death, and TUDCA has been shown to reduce neurodegeneration in neurological diseases, exerting neuroprotective and neurogenesis-promoting effects via mitochondrial regulation [55,56].

BAs can influence neuroplasticity by activating their receptors, affecting neuronal maturation, survival, and synaptic remodeling, ultimately alleviating depressive symptoms. Research has shown that FXR can affect the activity of the cAMP response element binding protein (CREB). CREB regulates brain-derived neurotrophic factor (BDNF) expression by binding to the promoter region of the BDNF gene. BDNF, in turn, can activate the CREB signaling pathway, promoting CREB phosphorylation and increased activity [57,58]. BDNF plays a critical role in neuron survival, synaptic connectivity, learning, and memory in the nervous system [59]. Depression patients often have lower BDNF/CREB levels in the brain [60], and some antidepressants have been found to increase their activity [61]. In the hippocampal CA1 region of depression model mice, FXR overexpression may lead to the cytoplasmic translocation of CREB-regulated transcription coactivator 2 (CRTC2), reducing CREB phosphorylation and activity, which in turn lowers BDNF expression [37,62]. An Alzheimer’s disease (AD) study also demonstrated that FXR overexpression interacts with CREB, decreasing CREB and BDNF protein levels, increasing amyloid-β (Aβ) deposition, and aggravating Aβ-induced neuronal apoptosis. In contrast, low FXR expression reversed these effects. Thus, the FXR-BDNF/CREB signaling pathway may represent a novel strategy for treating depression [63].

CUMS downregulates hippocampal LXR levels in mice, with LXRβ expression inversely correlating with depression-like behavioral severity. Hippocampal LXR knockdown induces depression-like phenotypes and impairs neurogenesis, whereas pharmacological activation of LXRα/β using agonist GW3965 rescues these behavioral deficits and reverses neurogenesis suppression. This mechanistic exploration combining genetic knockdown and pharmacological approaches establishes LXRβ’s critical role in depression pathogenesis, providing an experimental foundation for developing LXR-targeted therapeutic strategies [38].

TGR5 is expressed in multiple brain regions associated with memory, including the hippocampus and prefrontal cortex. The TGR5 agonist INT-777 significantly improved spatial memory impairments in mice [64], and is considered neuroprotective. By upregulating TGR5 in the lateral hypothalamus, inhibiting GABAergic neuron excitability, TGR5 can enhance excitatory transmission from CA3 pyramidal neurons in the hippocampus to the lateral septal nucleus, reducing immobility in the tail suspension and forced swim tests in socially defeated mice, increasing sucrose preference and social interaction behaviors [16,39]. INT-777 also promotes CREB phosphorylation, upregulates BDNF and synaptic protein expression in Alzheimer’s model mice, benefiting synaptic plasticity [46]. Another TGR5 agonist, UDCA, commonly used to treat cholestatic liver diseases and gallstones, was administered orally from olive leaves to high-fat diet model mice. The results showed increased BDNF mRNA levels in the hippocampus, weight loss, improved cognitive function and fewer depressive symptoms [65]. These findings suggest that UDCA may be a potential candidate for treating depression.

## 4. The Relationship Between Bile Acids, Their Receptors, and Anxiety

Anxiety and depression, as highly comorbid psychiatric disorders, share pathological underpinnings, including neurotransmitter imbalances and hypothalamic–pituitary–adrenal (HPA) axis hyperactivity [66]. Clinically, both conditions exhibit overlapping BAs dysregulation [25], with serum primary BAs (e.g., CDCA) showing inverse correlations with anxiety severity, while secondary BAs (e.g., LCA derivatives) are markedly elevated. Mechanistically, these metabolites may mediate anxiety by activating central FXR receptors, thereby disrupting neurotransmitter release and synaptic plasticity. Conversely, TGR5 agonists (e.g., TUDCA) ameliorate anxiety-like behaviors via dual mechanisms: suppressing neuroinflammatory cascades and enhancing BDNF-mediated synaptic resilience.

Animal studies have systematically delineated the pathogenic role of BAs metabolic disturbances. In chronic inflammatory pain models, fecal microbiota transplantation from socially subordinate mice exacerbated anxiety-like phenotypes in pseudo-germ-free recipients, with anxiety behaviors inversely correlating with elevated levels of specific BA metabolites (D8′-merulinic acid A, 12-ketodeoxycholic acid, 7-oxolithocholic acid, 3α-hydroxy-7-oxo-5β-cholanic acid). Metabolomic analyses linked these metabolites to gut microbiota taxa such as Coriobacteriaceae UCG-002 [67]. In high-fat/high-sugar diet models, obeticholic acid (OCA) ameliorated neuroinflammation by reducing hippocampal microglial infiltration and IL-1β expression while normalizing taurocholic acid levels [68]. Bile duct ligation-induced hepatic encephalopathy rats demonstrated that hepatic BA accumulation disrupted AMPK-mediated energy metabolism and neuroprotection, exacerbating anxiety, whereas gallic acid alleviated symptoms via AMPK activation [69]. Clinical parallels emerge in primary biliary cholangitis patients, where altered BA profiles and dysbiosis co-occur [70], suggesting gut microbiota-BA crosstalk in gut–brain signaling. Mendelian randomization by Xiao et al. [71] confirmed causal links between primary BA biosynthesis (e.g., CA metabolism) and anxiety risk, mediated by nuclear receptor–GPCR interplay. Notably, 12-ketodeoxycholic acid and 7-oxolithocholic acid inversely correlated with anxiety, supporting their biomarker potential.

Genetic knockout models further validated the roles of BA receptors. FXR-null mice exhibited reduced anxiety but impaired cognition and motor coordination, attributed to elevated hippocampal GABA/Glu ratios and cerebellar monoamine dysregulation (NE, 5-HIAA) [72]. Conversely, TGR5 deficiency increased anxiety-like behaviors, paralleling reduced hippocampal 5-HT and 5-HT1A receptor expression. Remarkably, transplanting TGR5−/− microbiota into wild-type mice recapitulated anxiety and serotonergic deficits [73], underscoring microbiota–gut–brain axis mediation.

Similar BA-anxiety associations extend to irritable bowel syndrome (IBS) and Crohn’s disease (CD). Aziz et al. [74] identified idiopathic bile acid diarrhea (BAD) in 25% of diarrhea-predominant IBS patients, where elevated colonic BAs drive visceral hypersensitivity and gut–brain axis dysregulation, amplifying anxiety risk. BouSaba et al. [75] reported higher anxiety/depression scores and BA sequestrant use in BAD-positive IBS patients. In CD, fecal HDCA and 12-DHCA inversely correlated with depression, while serum GCA and 7-DHCA positively associated with anxiety [76].

Collectively, these findings elucidate multidimensional mechanisms linking BA metabolism to anxiety via nuclear receptor signaling, microbial modulation, and gut–brain communication, providing a robust rationale for BA-targeted therapeutic strategies in mood disorders.

## 5. Therapeutic Applications and Clinical Challenges of Bile Acid-Targeted Agents

### 5.1. Dual Challenges in BA Metabolic Profiling and Pharmacokinetics

Substantial interindividual variability in gut microbiota composition, BA metabolic activity, and receptor expression/function contributes to heterogeneous therapeutic responses to BA-based interventions in depression, with additional modulatory effects from age, sex, genetics, and lifestyle factors [77]. Dynamic fluctuations in BA profiles, driven by microbial BA hydrolase (BSH)-mediated dehydroxylation/oxidation of primary BAs into secondary forms (e.g., DCA, LCA) [78,79], postprandial BA release rhythms, and disease-associated metabolic dysregulation (e.g., diabetes, hepatotoxicity) [80,81], pose significant challenges for reliable BAs quantification. Technical limitations further compound these issues: short-term sample stability (serum BAs degrade within 1 week at 4 °C and 2 months at −20 °C) [82], detection sensitivity constraints for low-abundance BAs and structurally complex microbial-conjugated BAs (MCBAs), and analytical complexity in disentangling confounders (diet, genetics, comorbidities) through multiomics integration [78].

Interspecies differences in BAs metabolism critically limit translational validity. Murine-specific muricholic acids (MCAs), which exhibit FXR-antagonistic properties absent in humans, confound preclinical findings. Cyp2c70-knockout models, generating human-like BA profiles by abolishing MCA synthesis, highlight the necessity of species-tailored mechanistic studies [83].

Pharmacokinetic hurdles include poor oral bioavailability (<3% for OCA) due to rapid bacterial hydrolysis, hepatic/renal clearance, and intestinal degradation of hydrophilic BAs (e.g., UDCA), versus membrane toxicity risks from hydrophobic analogs (e.g., DCA). Structural optimization strategies (6α-ethyl or sulfonate modifications) enhance metabolic stability and hepatobiliary secretion efficiency, exemplified by OCA’s taurine-conjugated derivative [84]. However, systemic exposure risks (bile acid accumulation-related cholestasis, diarrhea) and BBB penetration limit CNS-targeted efficacy. Innovative delivery systems (micelles, enzyme-responsive lipid nanoparticles [LNPs]) may overcome these barriers by improving solubility, tissue-specific release, and target organ enrichment [85,86].

### 5.2. Current Clinical Trial Landscape

Clinical trials investigating BA-based therapies for depression remain exploratory, with current evidence predominantly derived from preclinical models. However, the mechanistic roles of BAs in modulating gut microbiota metabolism and nuclear receptor signaling, coupled with established gut–brain axis dysregulation in depression, warrant further clinical validation of their therapeutic potential.

While BA-focused trials for neurological disorders are limited, recent studies in amyotrophic lateral sclerosis (ALS) highlight both promise and challenges. TUDCA, a hydrophilic BA with neuroprotective properties (e.g., mitochondrial stabilization, anti-apoptotic effects), initially showed potential in delaying functional decline and extending survival. However, Phase III trials failed to demonstrate significant efficacy versus placebo, though combination therapy with sodium phenylbutyrate exhibited transient benefits despite unmet secondary endpoints and frequent gastrointestinal adverse events [87].

Obeticholic acid (OCA), an FXR agonist FDA-approved for primary biliary cholangitis (PBC) and nonalcoholic steatohepatitis (NASH), achieved histological improvement (≥1-stage fibrosis reduction without NASH worsening) in 22.4% of patients at 25 mg (vs. 9.6% placebo) over 18 months, with enhanced efficacy in F3-stage subgroups (25.4% vs. 12.3%) [88,89,90]. Long-term safety data (>8000 patient-years) confirm tolerability, with pruritus (54.8%) as the primary adverse event. Notably, OCA improved liver enzymes (ALT, AST, GGT) and stiffness even in non-responders. However, 2024 FDA warnings identified severe hepatotoxicity risks (including the need for liver transplantation) in non-cirrhotic PBC patients, prompting revised monitoring protocols for liver function markers (ALT, bilirubin) and symptom vigilance (jaundice, hematemesis) [91].

FXR agonism may induce pruritus via BA accumulation-driven cutaneous nerve sensitization and dyslipidemia (elevated LDL-C, reduced HDL-C), potentially exacerbating cardiovascular risks in depression patients [92]. Additional safety concerns include gastrointestinal disturbances, hepatotoxicity, and neuropsychiatric effects, necessitating comprehensive toxicological profiling and drug interaction studies.

Regulatory pathways for BA therapies prioritize rigorous safety and efficacy validation, particularly for liver indications. While no specific guidelines exist for neuropsychiatric applications, evolving mechanistic insights into BA–CNS interactions may prompt future FDA policy adaptations. Translation to depression treatment requires standardized quality control, large-scale trials, and optimized dosing strategies to balance efficacy and tolerability.

## 6. Summary and Outlook

BAs modulate depression pathogenesis through FXR, LXRs, and TGR5 receptors, orchestrating neuroinflammation (via NF-κB/NLRP3 suppression), gut microbiota metabolism (via GLP-1/FGF19 regulation), and neural plasticity (via BDNF/CREB activation). Depressed patients exhibit a disrupted primary–secondary BA axis, with specific profiles (e.g., reduced CDCA, elevated LCA) correlating with symptom severity and treatment resistance, underscoring their potential as dynamic biomarkers. Notably, traditional Chinese medicine (TCM), including herbal therapies and acupuncture, may regulate cholesterol enterohepatic circulation and nuclear receptor pathways, offering favorable efficacy and safety profiles in depression management [93,94]. Notably, divergent BA concentration profiles are observed across depression models: serum GCA is elevated with concurrent CA reduction in CUMS rats, whereas restraint stress models exhibit significantly increased serum CA levels. These opposing outcomes may be attributed to inter-model variability (e.g., stress paradigms) or temporal sampling discrepancies [95].

Key BA–depression linkages span inflammatory, microbial, and neuroplastic domains. Neuroinflammation: TUDCA normalizes hippocampal and mPFC TNF-α/IL-6 levels in CUS mice. FXR deletion exacerbates hepatic inflammation, while ganoderic acid A ameliorates depressive behaviors via FXR modulation. LXRβ agonists suppress neuroinflammatory cascades. Gut microbiota: Severe depression correlates with altered BA concentrations (e.g., TLCA inversely linked to depression scores) and dysbiosis (e.g., enriched Coriobacteriaceae). CUMS mice show fecal DCA associations with Firmicutes; CSS restores microbial–BA homeostasis. Neural plasticity: Hippocampal FXR overexpression reduces BDNF in depressed mice, mirroring low BDNF/CREB levels in patients. CUMS downregulates LXRβ, impairing neurogenesis, reversible by LXRα/β agonists. TGR5 activation enhances synaptic proteins, while UDCA elevates hippocampal BDNF mRNA, alleviating depressive phenotypes.

Future research must elucidate BBB transport mechanisms, receptor spatiotemporal specificity, and bidirectional regulation (e.g., FXR overexpression’s neuroplasticity trade-offs). Innovations should prioritize BBB-penetrant delivery systems (nanocarriers) and structural optimization (sulfonate modifications) to enhance stability. Clinical translation requires validating BA-based therapies (e.g., TUDCA-antidepressant combinations) with emphasis on interindividual variability (microbiota, genetics) and long-term safety profiling (cardiometabolic/neuropsychiatric risks). By addressing these challenges through interdisciplinary collaboration, BA-centric strategies may redefine depression therapeutics.

## Figures and Tables

**Figure 1 ijms-26-04023-f001:**
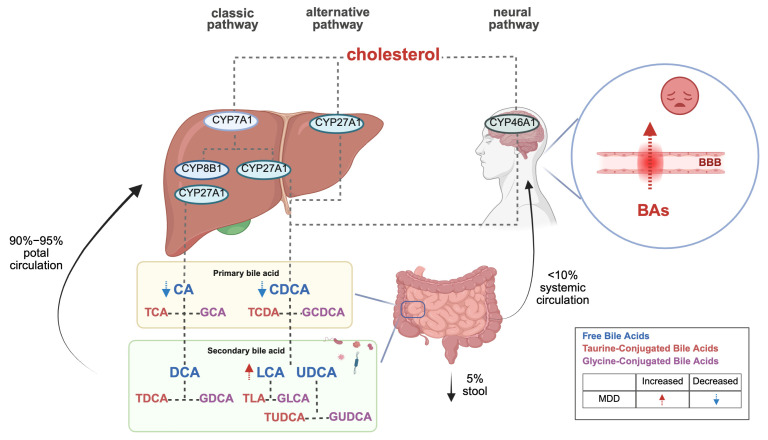
Bile acid biosynthesis pathways and their role in depression.

**Figure 2 ijms-26-04023-f002:**
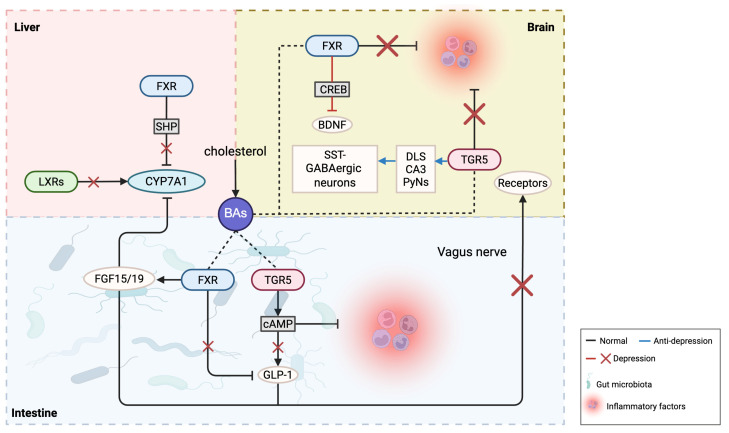
The possible roles of bile acids and their receptors in depression.

**Table 1 ijms-26-04023-t001:** A summary of the application of bile acids and their receptors in depression.

	BAs/BAR	Model		Intervention	Results	Refs
Inflammation	TUDCA	CUS	The CUS rats were exposed to two mild stressors per day for five weeks. Stressors: cage shaking for 1 h, lights on during the entire night, placement in 4 °C cold room for 1 h, mild restraint for 2 h, 45° cage tilt for 14 h, lights off for 3 h during the daylight phase, wet cage for 14 h, flashing light for 6 h, noise in the room for 3 h, and water deprivation for 12 h during the dark period	TUDCA pretreatment	↓Pro-inflammatory cytokines in the PFC and hippocampus (IL-6, TNF-α, IL-1β, ↓Iba-1, NLRP3)	[31]
		LPS	LPS (i.p. −0.83 mg/kg)		↓Pro-inflammatory cytokines in the PFC and hippocampus (IL-6, TNF-α)	[32]
	FXR	CSDS	For a total of 10 days, a single male C57BL/6J intruder mouse was exposed to a different male CD1 aggressor mouse for 10 min each day. Following 10 min of contact, the intruder C57BL/6J mouse was housed across a perforated iron gauze divider, providing further stressful sensory cues from the aggressor CD1 mouse for the remainder of the 24 h period. Control C57BL/6J mice were housed in pairs in defeat boxes with one mouse per side of the perforated divider. All control mice that were placed with the controls were changed daily	GAA	↑FXR in the PFC; ↓pro-inflammatory cytokines in the PFC (NLRP3, caspase-1, IL-1β)	[33]
	↓LXRβ in the basolateral amygdala	CUMS	The CUMS mice daily received 2 stressors in combination that were prior randomly scheduled for a 28 d period. Stressors: 24 h of food deprivation, 24 h of water deprivation, 1 h of exposure to empty bottles, 8 h of cage tilt (45°), overnight illumination, 24 h of habitation in a soiled cage (200 mL of water in 100 g of sawdust bedding), 30 min of forced swimming at 8 °C, 2 h of physical restraint, 24 h of exposure to a foreign object	LXRβ agonists: TO901317	↓NF-κB/NLRP3, IL-1β	[34]
		CORT	Mice received CORT drinking water daily for a 28 d period. CORT (Sigma–Aldrich, St. Louis, MO, USA) was dissolved in 100% ethanol (Sigma–Aldrich) and mixed with potable water at a final concentration of 0.1 mg/mL CORT and 1% ethanol			
	Abnormal activation of the LXR/RXR pathway	CUMS	/	/	Inflammatory response (TLR4 activation)	[35]
Gut Microbiota	↓HCA; 7-ketoDCA	CUMS	/	CSS	↑Intestinal *P. distasonis* abundance	[36]
	↑FXR in the hippocampal CA1 region	CUS	/	FXR shRNA	↑CREB-BDNF	[37]
Neuro-plasticity	↓LXRβ in the hippocampal	CUS	/	LXRα/β agonists: GW3965; LXRβ shRNA	↑Hippocampal neurogenesis	[38]
	↓TGR5 in the LHA	CSDS	/	TGR5 agonists: INT-777	Activation of the DLS CA3 PyNs →SST—GABAergic neurons pathway	[39]

Note. ↑, upregulated; ↓, downregulated.

## Data Availability

All data are reported in the text.

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
