# Peer review of "Research Progress on the Mechanism of Bile Acids and Their Receptors in Depression"

_ijms, 2025, doi:10.3390/ijms26094023_

Round 1
Reviewer 1 Report
Comments and Suggestions for Authors
This review give a comprehensive and well-structured discussion about the relationship between bile acids (BAs), their receptors (FXR and TGR5), and depression. The authors cover many important mechanisms, including inflammation, gut microbiota, neuroplasticity, and energy metabolism, and also discuss potential therapeutic applications.
I consider that the topic is highly relevant, because increasing evidence suggest that bile acid metabolism play a key role in mental health. The review widely discuss how BAs modulate inflammation, neuroplasticity, gut microbiota, and metabolic pathways related to depression.
Comments
While the review summarize many studies, it do not include results of quantitative comparisons or systematic analysis of the evidence. What about data on key bile acids (e.g., TUDCA, CDCA) and depression markers?
Authors should compare BA levels in depressed vs. healthy individuals to assess their diagnostic potential.
The ms rely too much on text but lack visual summaries. I considerar that adding a graphical summary showing how bile acids interact with FXR/TGR5 and influence depression could help.
Improve Fig. 1, which currently do not clearly connect BA synthesis with mental health.
Include a comparison in Table 1 showing different BA-based interventions in depression models.
The review mainly presents supportive studies, but conflicting findings are not discussed. Pleasee discuss variability in bile acid levels in different depression models.
Address challenges in measuring BAs accurately (e.g., variability in metabolomics techniques) shouldd be discussed.
The review mention BA-based treatments, but do not critically discuss their limitations.What about pharmacokinetics and bioavailability challenges of BA-based therapies?
Address potential side effects of FXR/TGR5 modulation in depression treatment.
If bile acid therapeutics are moving toward clinical use, regulatory challenges must be addressed. Is there any FDA regulation for BA-based drugs?
Compare current clinical trials involving FXR/TGR5 agonists in neurological disorders.
Minor Issues
Clarify Some Sections: The mechanistic descriptions of FXR/TGR5 signaling are dense—simplifying these will improve readability.
Please mprove Table 1 Include more details on the experimental models used in BA-depression studies, or include a new table
Some sentences are too complex and should be revised for clarity.
Reviewer 2 Report
Comments and Suggestions for Authors
Comments:
- Line 77, “with a minority entering systemic circulation” this sentence is quite confusing, please rephrase.
- Figure quality seems very poor, Fig 2, tough to even visualize, please enhance the quality of your figures with higher DPI/resolution.
- In vivo and in vitro, should be in italics.
- As Liver X Receptors (LXRs) also play a critical role in regulating bile acid production. Did author observe any link between LXR, bile acids and depression?
- I am curious to know If there is any effect of bile acids production and its receptors on anxiety disorder too, as anxiety and depression are often related mental health disorders. I suggest adding a brief paragraph on BAs and anxiety disorders that will strengthen your study on bile acids and other mental disorders along with depression which is primary focus.
- Table 1, Inflammation, gut microbiota and neuroplasticity should be labeled on the top for consistency, just like other parameters.
- 1, line 188, inflammatory should be change to inflammation.
- Line 196-198, “Compared with normal people, serum inflammatory factors (NLRP3, IL-18, NF-κB) in major depressive disorder (MDD) patients were remarkably higher” this sentence doesn’t make any sense to me, please rephrase.

Can be improved!
Round 2
Reviewer 1 Report
Comments and Suggestions for Authors
The authors have satisfactorily addressed the queries and have enhanced their article accordingly
Author Response
We sincerely thank the reviewer for the constructive feedback, which has greatly improved our manuscript.
Reviewer 2 Report
Comments and Suggestions for Authors
Just a minor suggestion
Comments:
- Fig 1 & 2, resolution of small boxes on the right still needs to be increase as its very low in font size.
- In abstract, through you have defined LCA and CDCA in later sections, but this must be spell put when used the term 1st Same with other receptors.
I have no further comments.

Author Response
Comments 1:Fig 1 & 2, resolution of small boxes on the right still needs to be increase as its very low in font size.
Response 1:Thank you for pointing this out.Have been modified.
Comments 2:In abstract, through you have defined LCA and CDCA in later sections, but this must be spell put when used the term 1st Same with other receptors.
Response 2:Have been modified.